# Prognostic Potential of Cancer-Associated Fibroblast Surface Markers and Their Specific DNA Methylation in Prostate Cancer

**DOI:** 10.3390/diagnostics15192434

**Published:** 2025-09-24

**Authors:** Mark Jain, Olga Nesterova, Mikhail Varentsov, Nina Oleynikova, Aleksandra Vasiukova, Sofia Navruzova, German Golubin, Larisa Samokhodskaya, Pavel Malkov, Armais Kamalov

**Affiliations:** 1University Clinic, Lomonosov Moscow State University, 119992 Moscow, Russia; oy.nesterova@gmail.com (O.N.); noleynikova@mc.msu.ru (N.O.); slm61@mail.ru (L.S.); malkovp@yandex.ru (P.M.); priemnaya@mc.msu.ru (A.K.); 2Faculty of Fundamental Medicine, Lomonosov Moscow State University, 119992 Moscow, Russia; miha78st@gmail.com (M.V.); aleksandraparshina17@gmail.com (A.V.); sofiakharcheva@gmail.com (S.N.); german.golubin@gmail.com (G.G.)

**Keywords:** prostate cancer, cancer-associated fibroblasts, tumor microenvironment, prognosis, prognostic biomarker, biochemical recurrence, DNA methylation

## Abstract

**Background:** Cancer-associated fibroblasts (CAFs) are a key component of the prostate cancer (PCa) microenvironment, the abundance of which is often linked to poor prognosis. The surface markers for CAFs are mostly established, yet our current knowledge of epigenetic alterations in them remains limited. The aim of this study was to evaluate the relationship between CAF-specific DNA methylation, their abundance and the PCa prognosis. **Methods:** The study included 88 PCa patients with known presence or absence of a biochemical recurrence within a 6-year period. Resected PCa tissue was assessed for the surface expression of FAP, PDGFRb, CD90, and POST, and for the methylation of *EDARADD*, *GATA6*, and *PITX2* genes using qPCR and ddPCR. **Results:** The surface expression of FAP, PDGFRb and CD90 was associated with a higher Gleason score (*p* < 0.05). The analytical sensitivity of ddPCR was superior to qPCR; results obtained using ddPCR demonstrated a more significant association with clinical features of PCa. *EDARADD* methylation and PDGFRb expression were associated with a risk of biochemical recurrence (HR–0.961 [95% CI: 0.931–0.991] and HR–2.313 [95% CI: 1.054–5.088]; *p* < 0.05, respectively). **Conclusions:** Upon further validation, the abundance of CAFs and their specific methylation might become a promising tool for the assessment of prognosis in PCa after radical treatment.

## 1. Introduction

According to the latest report of Global Cancer Statistics (GLOBOCAN), prostate cancer (PCa) ranks 4th in incidence and 8th in mortality, and it ranks 1st in both for males [1]. It is estimated that by 2040 the number of new cases of PCa will double and exceed 2.8 million annually [2]. This trend is associated, among other factors, with the increase in life expectancy of the population, which inevitably entails an increasing burden on the healthcare system and requires the introduction of new tools for diagnosis and risk stratification to guide treatment decisions [3].

PCa is a disease that largely depends on a personalized approach to diagnosis and treatment, especially in the localized form, where the choice of tactics may vary between an active surveillance and a radical treatment program [4]. Risk stratification is based on a combined assessment of prostate-specific antigen (PSA) levels, Gleason score and clinical stage, followed by the use of other clinicopathological tools: PSA kinetics, nomograms (Briganti, Partin, and Memorial Sloan Kettering Cancer Center (MSKCC)), and assessment of histological variants [5,6]. Taken together, these tools are useful, but in real clinical practice they are often insufficient to reliably predict the course of the disease and assess its aggressiveness [7]. Thus, in recent decades, there has been an active search for new prognostic biomarkers (circulating tumor nucleic acids, exosomes, epigenetic alterations, surface protein expression, proteomic and transcriptomic signatures, etc.) related not only to the tumor itself but also to its microenvironment (TME) [8,9].

Recent studies indicate that the TME represented by both cellular and acellular components is among the key factors responsible for the course of most oncological diseases, including PCa [10,11,12]. Growing evidence suggests that it is heavily involved in the multistep processes of tumor growth, metastasis formation, and response to therapies [13,14]. Due to both the development of sensitive single-cell analytical techniques and the advances in the molecular characterization of tumor tissue, a key component of the TME was identified, namely cancer-associated fibroblasts (CAFs) [15]. This term stands for a diverse population of fibroblast-like cells, which orchestrate multicellular stromal-dependent alterations and contribute to the above-mentioned tumor-related processes [14].

Generally, analysis of CAFs in tumor tissue specimens is impaired by their heterogeneity, although it is known that they may be distinguished from normal prostate fibroblasts (NPFs) based on the surface expression of certain proteins, such as fibroblast activation protein (FAP), platelet-derived growth factor receptors β (PDGFRb), cluster of differentiation 90 (CD90, also known as thymocyte differentiation antigen 1), and periostin (POST) [16]. At the genetic level, their identification is also difficult due to the fact that, unlike malignant cells, they are not characterized by a high frequency of mutations, as their cancer-associated phenotype is explained primarily by epigenetic changes [17,18]. A recent single cell methylome study revealed that in PCa CAFs could be distinguished from NPFs based on the methylation of certain genes, such as *PITX2* (hypomethylation) and *GATA6* (hypermethylation), whereas hypomethylation of the *EDARADD* gene appeared to be significantly associated with a poor prognosis [19]. Therefore, analysis of these epigenetic alterations might emerge as a new method to assess both the CAF content and the prognosis of the disease, although their association with these parameters needs to be proven in settings reflecting the possible future clinical use scenarios, namely in PCa whole tissue specimens.

The aim of our study was to evaluate the associations between the above-mentioned epigenetic biomarkers, the immunohistochemical characteristics of the TME, and the prognosis in PCa patients. As the overwhelming majority of studies dedicated to the analysis of methylation in PCa for prognostic purposes are based on the use of the quantitative polymerase chain reaction (qPCR) [20], this study also sought to compare its performance to the digital droplet PCR (ddPCR), which is known for its resilience to certain PCR inhibitors and its ability to precisely quantify DNA even in cases with extremely low target fractions [21,22].

## 2. Materials and Methods

### 2.1. General Information

The study protocol was approved by the institution’s Local Ethics Committee (#3/17 by 17 April 2017), and it was conducted in accordance with the Declaration of Helsinki. All study participants signed informed consent forms. Patient enrollment was conducted in the Moscow State University Clinic from January 2018 to December 2021. It included 88 patients with PCa (ICD-10 C61). The diagnosis was verified at a histological examination of the resected prostate tissue. The outcome of the disease (presence of a biochemical recurrence) was assessed in March 2024. The demographic and clinical characteristics of the study participants are presented in Table 1, whereas more detailed data are available in Appendix A.

### 2.2. Immunohistochemical Analysis

The study material consisted of resected PCa tissue. An immunohistochemical analysis was performed to investigate CAFs using rabbit antibodies directed against the antigens FAP (clone SP325, 1:75; Abcam, Ltd., Cambridge, UK), PDGFRb (clone Y92, 1:100; Abcam, Ltd., Cambridge, UK), POST (1:120; Elabscience, Inc., Houston, TX, USA), CD90 (clone EP56, 1:100; Epitomics Inc., Cambridge, UK). Preprocessing (deparaffinization, rehydration, and antigen retrieval) was performed in a PT-Module (Thermo Fisher Scientific, Inc., Waltham, MA, USA) for 20 min (95–98 °C) at pH 8.0. The immunohistochemical reactions were conducted in a semiautomatic mode using the Autostainer 480S (Thermo Fisher Scientific, Inc., Waltham, MA, USA) with the PrimeBiomed detection system and DAB-chromogen (PrimeBiomed, Moscow, Russia).

In prostate cancer, PDGFRb was evaluated around tumor complexes. The intensity of the reaction was assessed in the designated area using a scale of 0 to 3 (0—no reaction, 1—weak reaction, 2—moderate reaction, 3—strong reaction). FAP and POST were evaluated using the same scale. CD90 showed a very heterogeneous and mosaic pattern, so this reaction was evaluated as a percentage of the stroma. Subsequently, the percentage was converted to a standard system (0–3) by analogy with other CAF markers as follows: 0–6%—negative; 7–15%—1, 16–39—2, >40%—3. The average staining intensities from 5 fields of vision were counted for each sample. The quantitative assessment of the stained micrographs was performed independently by two pathologists to achieve a more objective value. In case of discrepancy between observations, the mean value was calculated.

### 2.3. DNA Isolation and Processing

Formalin-fixed, paraffin-embedded postresection tumor tissue slides for each study participant (9 samples were not included due to insufficient material) were requested for the DNA methylation analysis (6 µm thick). DNA was isolated from the slides using the ExtractDNA FFPE kit (Evrogen, Inc., Moscow, Russia) according to the manufacturer’s instructions. The elution volume was set to 60 μL. DNA samples were frozen at −80 °C shortly after the extraction. DNA was bisulfite-converted using the BisQuick kit (Evrogen, Inc., Moscow, Russia) according to the manufacturer’s instructions. To evaluate the quality of both extracted DNA and bisulfite-converted DNA, a spectrophotometric analysis was performed using NanoDrop 2000 instrument (Thermo Fisher Scientific, Inc., Waltham, MA, USA). The following quality control criteria were applied: DNA concentration >10 ng/μL and DNA purity (A260/A280) 1.7–1.9. To assess the completeness of the bisulfite conversion, a set of qPCR reactions using oligonucleotides designed to amplify a specific non-converted fragment of the *ACTB* gene (Appendix A) was carried out.

### 2.4. Methylation-Specific Polymerase Chain Reaction

The methodology included methylation analysis of *PITX2*, *EDARADD*, and *GATA6* genes using both ddPCR and qPCR. The *ACTB* gene was selected as an internal control. The generation and droplets analysis for ddPCR were performed using the QX200 AutoDG ddPCR system (Bio-Rad Laboratories, Inc., Hercules, CA, USA). The Veriti 96-Well Thermal Cycler (Life Technologies Corporation, Carlsbad, CA, USA) was used for the amplification of the reaction mixture. All ddPCR procedures were performed according to the manufacturer’s instructions. The sequences of oligonucleotides used in the methylation-specific ddPCR and qPCR are available in Appendix A.

The following thermocycling protocol was used for ddPCR (*PITX2* and *ACTB*): incubation at 95 °C (10 min), 45 cycles of denaturation at 94 °C (30 s) annealing/extension at 60 °C, and incubation at 98 °C (10 min). The total volume of the reaction mixture was 22 μL. It included 11 μL of ddPCR Supermix for Probes (Bio-Rad Laboratories, Inc., Hercules, CA, USA), 0.9 μM primers, 0.25 μM probes, and 2 μL of bisulfite-converted DNA. The conditions for *EDARADD* and *GATA6* slightly differed due to the implementation of the EvaGreen^®^ chemistry: incubation at 95 °C (5 min), 40 cycles of denaturation at 95 °C (30 s) annealing/extension at 60 °C, incubation at 4 °C (5 min) and 90 °C (5 min). The total volume of the reaction mixture was 22 μL as well. However, it included 11 μL of ddPCR EvaGreen Supermix (Bio-Rad Laboratories, Inc., Hercules, CA, USA), 0.1 μM primers, and 2 μL of bisulfite-converted DNA.

For methylation analysis using qPCR, amplification was carried out on the CFX96 Touch System (Bio-Rad Laboratories, Inc., CA, USA) using the following protocol for *PITX2* and *ACTB*: incubation at 95 °C (5 min), 45 cycles of denaturation at 95 °C (15 s), annealing/extension at 60 °C (30 s). The total volume of the reaction mixture was 20 μL. It included 4 μL of qPCRmix-HS (Evrogen, Inc., Moscow, Russia), 0.4 μM primers, 0.2 μM probes, and 2 μL of bisulfite-converted DNA. The protocol for *EDARADD* and *GATA6* slightly differed as well due to the use of the SYBR Green chemistry: incubation at 95 °C (5 min), 40 cycles of denaturation at 95 °C (15 s), annealing at 60 °C (15 s), and extension at 72 °C (30 s). The total volume of the reaction mixture was 20 μL. It included 4 μL of qPCRmix-HS-SYBR (Evrogen, Inc., Moscow, Russia), 0.4 μM primers, and 2 μL of bisulfite-converted DNA.

Analytical characteristics of the assays were determined in a series of experiments using Qiagen control methylated and unmethylated DNA (Qiagen GmbH, Hilden, Germany. Methylated DNA was added in fractions ranging from 0% to 100%. The qPCR assays detected methylation with limits of 10–15%, while the ddPCR assays achieved a detection limit below 1%. Samples were analyzed in a single measurement. Reactions were repeated for samples that did not meet quality criteria. The quality criteria for qPCR included an *ACTB* (internal control) fluorescence threshold cycle of ≤32; the ddPCR quality criteria included the generation of >10,000 droplets per well and an *ACTB* level of >100 copies per well. The batch variability was controlled by ensuring that results from a positive control sample—containing an equal mixture of Qiagen control methylated and unmethylated DNA (Qiagen GmbH, Hilden, Germany)—deviated by no more than 10%. False-positive signals were assessed and subtracted from the results of the corresponding samples based on the threshold cycle or the number of methylation-positive droplets detected in a sample containing Qiagen control unmethylated DNA (Qiagen GmbH, Hilden, Germany).

Due to the significant presence of the “rain-effect” [23] in ddPCR wells containing primers for the unmethylated portion of the *EDARADD* gene and for methylated portion of the *GATA6* gene, these wells were ignored for the purposes of DNA methylation calculation, and the results were normalized based on the level of *ACTB* gene. DNA methylation in ddPCR experiments was calculated using the following formula:methDNAddPCR(%) = [methylated target gene level (copies/μL)/*ACTB* gene level (copies/μL)] × 100(1)

Note that for *GATA6* methylated target gene concentration was calculated as unmethylated copies of *GATA6* subtracted from copies of *ACTB*.

DNA methylation (%) for *PITX2* gene measured by qPCR was calculated using the following formula:methDNAqPCR(%) = 2^−ΔΔCt^ × 100(2)

DNA methylation (%) for *EDARADD* and *GATA6* were calculated using the following formula:methDNAqPCR(%) = 2^−ΔΔCt^/(2^−ΔΔCt^ + 1) × 100(3)ΔΔCt = ΔCtsample − ΔCtcontrol(4)ΔCt = Ctmethylated − Ctunmethylated(5)

“Ct” denotes the threshold fluorescence cycle; “control” is a sample containing an equal mixture of Qiagen control methylated and unmethylated DNA (Qiagen GmbH, Hilden, Germany); “sample” is the analyzed sample with an unknown methylation status.

### 2.5. Statistical Analysis

All statistical analyses were performed using R software (R version 2025.05.1+513). Baseline nominal variables in independent samples were compared using the Chi-square test, Chi-Square Test with Yates’ Correction and Fisher’s exact test, while quantitative variables—using the Mann–Whitney test. Baseline quantitative variables in dependent samples were compared using Wilcoxon rank test. For correlation analysis we used Spearman rank correlation. Quantitative variables are presented as median and quartiles 1 and 3 (Q1–Q3), qualitative variables—as absolute numbers (*n*) and percents (%).

Survival differences among different risk stratification groups were compared using the log-rank test and Kaplan–Meier plots. Univariate Cox regression analysis was employed to determine the relationship between clinical and pathological variables and prognosis. Forest plot was used for univariate Cox regression results. Subsequently, multivariate Cox regression analysis was conducted to identify independent prognostic factors for prostate cancer recurrence. Nomograms were constructed based on these independent prognostic variables.

The concordance index (C-index), receiver operating characteristic (ROC) with area under the curve (AUC) values were calculated for every timepoint with time-dependent ROC-AUC plot for different multivariate models. The C-indices were used to evaluate the discriminative ability and predictive efficiency of the models. The C-index varies between 0.5, which is a totally random model, and 1.0, meaning perfect predictions. Bootstrap was used for the internal validation of the developed models. *p* < 0.05 was defined as indicating a statistically significant difference.

## 3. Results

### 3.1. CAF Markers: Immunohistochemistry

The distribution of the CAF markers in the samples according to the proposed intensity scale is presented in Table 2. Representative examples of immunohistochemical analyses results are shown in Figure 1 and Figure 2.

The PDGFRb reaction in the prostate gland was evaluated in the stroma around the tumorous glands, as well as (for comparison) in the intact stroma around the glands. It was noted that the PDGFRb reaction was also present around the non-tumorous glands, especially around the large ducts and the zone of basal cell hyperplasia (Figure 1a,b). No reaction was observed around the atrophic glands or in the periurethral zone. The FAP reaction was mostly absent outside the tumorous complexes, except for the large ducts that contained secretions.

In most cases (60/88), POST reaction was negative around intact glands (Figure 2c). In 16/88 cases, there was a mosaic-like weak reaction, and in 12/88 cases, the reaction was clearer (Figure 2d), but the POST reaction was always more pronounced around tumor complexes (Figure 2c,d). The POST reaction was also observed in the walls of large vessels. CD90 was found to be expressed in nerve trunks (Figure 2e), ganglion cells, and small capillaries, including those that were spared (which is not surprising, as CD90 is expressed in endothelial cells according to the literature [24]). CD90 was not expressed around intact tumor glands (Figure 2e,f).

For the convenience of further statistical analysis (mainly for tests requiring binary data input), we divided the PCa samples into groups (separate for each CAF marker) based on the expression of the studied CAF markers (Table 3). To begin, samples were divided into groups with (*n* = 16) and without the FAP expression (*n* = 18). Next, into groups with no/mild/moderate (*n* = 57, low expression group) and strong PDGFRb expression (*n* = 31, high expression group). Furthermore, into groups with no/mild (*n* = 31, low expression group) and moderate/severe POST expression (*n* = 57, high expression group). Finally, into groups with no/mild (*n* = 66, low expression group) and moderate/severe CD90 expression (*n* = 20, high expression group).

### 3.2. Association of CAF Markers with Clinical and Morphological Features of Prostate Cancer

To determine relationships between different CAF markers we performed a correlation analysis (Figure 3). It was observed that there was a moderate positive correlation between FAP and PDGFRb expression (r = 0.544; *p* < 0.001) and a weak positive correlation between PDGFRb and CD90 (r = 0.271; *p* = 0.012).

A detailed comparison of main clinical and morphological features among patients with low and high CAF markers is presented in Appendix A, whereas a comparison of CAF markers expression depending on different clinical and morphological data is presented in Appendix A. It appeared that patients with high PDGFRb expression, high CD90 expression, and presence of FAP expression exhibited a higher Gleason score (Figure 4a,b,e–h). At the same time high CD90 expression was associated with a higher total PSA level (Figure 4d), whereas the presence of FAP expression—with a higher perilymphatic invasion (Figure 4c).

### 3.3. Association of DNA Methylation Profiles with Clinical and Morphological Features

We performed the assessment of DNA methylation profiles using two methods—qPCR and ddPCR. There were significant differences in the *PITX2* methylation levels in the data obtained using qPCR and ddPCR (0.0 (0.0–0.0) vs. 4.6 (1.7–8.3), *p* < 0.001, respectively, Figure 5a), the same was true for the *GATA6* methylation levels (82.3 (70.8–89.7) vs. 74.3 (45.3–82.1), *p* < 0.001, respectively, Figure 5c). However, there were no statistically significant differences for the *EDARADD* methylation levels (85.5 (74.4–93.5) vs. 78.3 (59.0–100.0), *p* > 0.05, respectively, (Figure 5b).

Statistical analysis of the data obtained using qPCR in relation to relevant clinical and morphological characteristics of patients revealed that there were no significant results (except pT stage: patients with pT3 stage had slightly higher methylation levels of *PITX2* than the rest). Detailed results regarding the above-mentioned analysis are available in Appendix A. Consequently, we shifted the focus of the study to analyzing the results obtained using ddPCR.

To establish a relationship between DNA methylation profiles in different genes we performed correlation analysis (Figure 6). It was observed that there was a positive correlation between *GATA6* and *PITX2* methylation levels (r = 0.527; *p* < 0,001); a negative correlation between *EDARADD* and *PITX2* (r = −0.485; *p* < 0.001); a negative correlation between *EDARADD* and *GATA6* (r = −0.741; *p* < 0.001).

The median methylation level of *PITX2* was 4.63 (1.69–8.31) %, mean—6.17 ± 5.62%, ranging from 0 to 27.5%. The median methylation level of *EDARADD* was 78.3% (59.0–100), mean—77.9 ± 21.6%, ranging from 26.2 to 100%. The median methylation level of *GATA6* was 74.3% (45.3–82.1), mean—61.4 ± 28.6%, ranging from 0 to 93.3%. For further analysis we divided patients with PCa according to median methylation levels of *PITX2*, *EDARADD* and *GATA6*. Patients with methylation levels less than the median formed a group with low DNA methylation. Patients with methylation levels at the median and above formed a group with high DNA methylation. A detailed comparison of main clinical and morphological features among patients with low and high *PITX2*, *EDARADD* and *GATA6* methylation levels is presented in Appendix A, whereas a comparison of methylation levels depending on different clinical and morphological data is presented in Appendix A.

Patients with high *PITX2* methylation levels exhibited a higher MRI lesion rate in prostate than those with low *PITX2* methylation levels (97.2% and 75.0%, respectively, *p* = 0.025) (Figure 7a). Regarding the histological characteristics of the PCa, Gleason score and pT stage were also elevated in patients with high *PITX2* methylation levels (Figure 7b,c). However, *EDARADD* and *GATA6* methylation levels were not associated significantly with the studied clinical and morphological features of PCa (*p* > 0.05).

It appeared that by most part the DNA methylation levels were not associated with the expression of CAF markers assessed using immunohistochemical analysis (*p* > 0.05, Appendix A). The only statistically significant relationship was found for *PITX2* and PDGFRb: patients with high PDGFRb expression were characterized by higher levels of *PITX2* methylation (7.5 (3.5–11.6) vs. 3.8 (1.5–6.7) %, *p* = 0.027).

### 3.4. Prediction of Prostate Cancer Recurrence

Of the 69 patients with known PCa outcomes, 15 patients (22.1%) experienced a recurrence. The mean time to recurrence was 24.1 months (median 25 months). A comparison of clinical and morphological parameters, DNA methylation levels, and CAF markers between patients with and without PCa recurrence revealed that patients with this outcome were characterized by higher levels of total PSA, higher Gleason scores, more frequent pT3 and pN stages, and perilymphatic invasion, *EDARADD* hypermethylation, lower expression of PDGFRb and higher expression of CD90 at immunohistochemical analysis (*p* < 0.05, Table 4). A graphical presentation for the statistically significant variables in patients with and without PCa recurrence is available in Figure 8.

Univariate Cox regression analysis identified eight clinicopathological characteristics associated with PCa recurrence: total PSA, Gleason score, pathologic T stage, pathologic N stage, perilymphatic invasion, *EDARADD* methylation level (as both continuous and categorical variables), PDGFRb expression (as both continuous and categorical variables), CD90 expression (as a categorical variable) (Table 5, Figure 9h).

The statistically significant categorical predictors according to univariate analysis were supported by Kaplan–Meier curves and the log-rank test to visualize the prognostic relevance of these predictors in each subgroup (Figure 9). It was observed that patients with a higher Gleason score, higher pT stage and pN stage, presence of perilymphatic invasion, higher PDGFRb and CD90 expression, and lower *EDARADD* methylation levels had a significantly worse prognosis (*p* < 0.05).

For further multivariable Cox regression analyses we developed three models: *Model 1* with standard clinical and morphological predictors, *Model 2* with new morphogenetic predictors and the mixed *Model 3*. During the development of these models, we performed a comparative analysis of all possible combinations of predictors and selected the most optimal model with the lowest Akaike information criterion value. *Model 1* (Table 5) included Gleason score and pN stage as independent predictors of PCa recurrence, as higher Gleason score and presence of pathological N1 stage were associated with higher risk of PCa recurrence. *Model 2* (Table 5) included *EDARADD* methylation level (as a continuous variable) and PDGFRb expression (as a continuous variable). The increase in PDGFRb expression level on each point increased the risk of PCa recurrence by 2.571 times (HR = 2.571; 95% CI: 1.340–4.935; *p* = 0.005). By contrast, the increase in *EDARADD* methylation level on each 1% decreased the risk of PCa recurrence by 1.038 times or by 3.7% (HR = 0.963; 95% CI: 0.939–0.987; *p* = 0.003). The mixed model (*Model 3*) included *EDARADD* methylation level, PDGFRb expression, and Gleason score (Table 6). It was shown that Gleason score 4 + 3 = 7 and more was associated with 6.247 times higher risk of PCa recurrence (HR = 6.247; 95% CI: 1.627–23.988; *p* = 0.008). The increase in PDGFRb expression level on each point increased the risk of PCa recurrence by 2.313 times (HR = 2.313; 95% CI: 1.054–5.088; *p* = 0.036). Whereas the increase in *EDARADD* methylation level on each 1% decreased the risk of PCa recurrence by 1.040 times or by 3.9% (HR = 0.961; 95% CI: 0.931–0.991; *p* = 0.013).

The time-dependent ROC analysis demonstrated that *Model 3* exhibited superior accuracy for recurrence-free survival (Figure 10b). In particular, the AUC values for predicting recurrence-free survival in *Model 3* at 1, 3 and 5 years were 0.777 (95% CI: 0.644–0.909), 0.911 (95% CI: 0.817–1.000), and 1.000 (95% CI: 1.000–1.000), respectively, while for *Model 1* and *Model 2* AUC values for each timepoint were lower (Appendix A). The same results were obtained using Harrell’s concordance index comparison (Figure 10c). C-indices for *Model 3* at 1, 3 and 5 years were 0.855, 0.878 and 0.876, respectively, while for *Model 1* and *Model 2* C-indices were lower for each timepoint as well (Appendix A). Based on the developed *Model 3*, we created a nomogram to assess the recurrence-free survival of patients with PCa at 1, 3, and 5 years after surgery, which proved the significance of the included factors (Figure 10a). Additionally, *Model 3* was validated internally using bootstrap (1000 replications): the Dxy was 0.727, whereas Harrell’s C-index was 0.864, confirming the discrimination ability of the proposed model.

## 4. Discussion

CAFs are a key component of TME, which is closely related to various clinical and morphological features of oncological diseases. For example, in ovarian cancer, it was shown that the presence of CAFs in the stroma is associated with a higher rate of metastatic lesions of lymph nodes and omentum [25,26]. Similar results were obtained for nasopharyngeal carcinoma, breast cancer, cholangiocarcinoma, and colorectal cancer [27,28,29]. Our study described the relationship between clinical and morphological features of PCa and several known surface markers of CAFs (FAP, PDGFRb, CD90 and POST). It was observed that FAP, PDGFRb, and CD90 are associated with a less favorable profile of clinical and morphological features of PCa: a higher PSA level before surgery, higher probability of lymph node damage, and higher Gleason tumor grade. These results confirm that the abundance of CAFs may be associated with more aggressive morphological manifestations of PCa.

Similar results were obtained in a study by Wu et al., where CAFs were detected using FAP and α-smooth muscle actin (α-SMA) surface markers [30]. It was demonstrated that the expression of these markers is present in stromal fibroblasts and is absent in tumor cells. Additionally, it was found that FAP and α-SMA are significantly more common in the tumor stroma compared to the stroma of the normal prostate gland. Assessment of the relationship of the FAP expression with clinical and morphological features of PCa revealed that in the presence of FAP, stage cT3 and higher occurred in 61.1% of patients, while in the absence of FAP—only in 42.1% (*p* = 0.003). Further analysis allowed the authors to establish a clear relationship between FAP and Gleason score. In 87.0% of patients with detectable FAP in the stroma, the Gleason score was 7 or higher (*p* = 0.031) [30].

In our cohort, a different CAF surface marker, namely PDGFRb, was also associated with unfavorable clinical and morphological features of prostate cancer, in particular with Gleason score. Similar association was demonstrated in a recent study by Ageeli et al., where researchers evaluated the relationship of CAF surface markers with Gleason score and prostate tissue stiffness based on ultrasound analysis results: an increase in Gleason score from 3 + 3 to 4 + 3 was accompanied by a 10% increase in the expression of PDGFRb [10].

The adverse effect of CAFs on clinical and morphological features of PCa was observed in animal models and cell cultures as well. Linxweiler et al. demonstrated that the inoculation of mice with CAFs and tumor cells leads to an increase in PSA levels and to a higher metastatic lesion rate in the lymph nodes and lungs compared to mice inoculated with tumor cells only or tumor cells with NPFs [31]. Shen et al. showed that co-cultivation of CAFs with PCa tumor cells leads to an increased perineural invasion [32].

The possible mechanism by which CAFs promote the progression of PCa might be an increase in proliferation, survival and migration of tumor cells as well as a local immunosuppression under the influence of cytokines secreted by CAFs. Sun et al. demonstrated that the concentration of TGF-β1 in a conditioned environment of tumor cells with CAFs was significantly elevated compared to that of isolated tumor cells. Also, it was noted that an inhibitor of the TGF-β1 receptor could suppress CAF-induced cell proliferation and migration of PCa cells [33]. It is known that CAFs can secret other mediators which promote immunosuppression: Programmed cell death 1 (PD-1) and its ligand (PD-L1), vascular endothelial growth factor, interleukins 6, 8, 23 [34]. Moreover, it was shown that CAFs are able to increase the expression of several transcription factors and the activity of several pathways involved in PCa cell proliferation, thereby enhancing the invasive potential, such as YAP1 and Eph/Ephrin, Notch, MAPK pathways [35,36,37,38].

DNA methylation is an essential epigenetic regulatory event that plays a key role in tumorigenesis by altering the behavior of cells comprising both the tumor and its microenvironment [39]. Many studies focused on the identification and examination of methylation-based prognostic biomarkers in PCa, yet the data on the epigenetic regulation of CAFs for this disease is quite limited [20,40,41,42]. This study sought to investigate the prognostic utility of three epigenetic biomarkers which were recently revealed in a single-cell methylome analysis of prostate CAFs: methylated copies of *PITX2* and unmethylated copies of *GATA6* were expected to reflect the general CAF content in the sample, whereas unmethylated copies of *EDARADD*—the aggressiveness of these cells [19]. Multivariate logistic regression allowed to account for the cellular abundance of CAFs due to the inclusion of the results immunohistochemical analysis of tissue samples for known CAF markers into the models.

In this study, DNA methylation was assessed using two different techniques—methylation-specific qPCR and ddPCR. It appeared that the latter was superior in terms of both the analytical sensitivity and the association of the results with relevant clinical variables. This observation is in line with findings of various other studies dedicated to the comparison of these two techniques for the purposes of methylation analysis [43,44,45]. However, these results do not undermine the known effectiveness and convenience of methylation-specific qPCR, as this method was proven to be a reliable approach for the analysis of FFPE samples in PCa in numerous studies [20,46,47]. The advantages of ddPCR might be outweighed by the prices of the instruments, reagents, and consumables. ddPCR systems are 5–10 more expensive than those for qPCR, and the price of consumables per reaction is higher by approximately the same margin, limiting the availability of these systems across pathology laboratories. The medico-economic benefit of its translation into clinical practice for the purposes of the prognosis evaluation in PCa needs to be investigated further.

Among the biomarkers analyzed in this study, *PITX2* methylation stands out from the rest as its positive association with a negative prognosis in PCa has been known for quite some time [48,49]. In our cohort, methylation of *PITX2* was not significantly associated with the recurrence of the disease. It is worth noting that the prognostic relevance of this biomarker was not studied previously in the population of our region, and PCa is known to have significant regional and population-level genetic differences, which might explain this discrepancy [50]. However, the level of methylated *PITX2* positively correlated with the expression of PDGFRb, which was in turn the only CAF marker associated with the prognosis, whereas patients with hypermethylated *PITX2* were characterized by a higher MRI lesion rate, Gleason score, and pT stage, which to some extent prove the prognostic significance of this epigenetic alteration. These findings contradict the assumption that hypomethylation of *PITX2* might reflect the general CAF content in a sample. It is likely that the abundance of methylated copies of this gene stems from a different cellular source and, thus, obscures the contribution of DNA from CAFs. Interestingly, single cell methylome analysis of PCa, compared to a matching benign-appearing tissue, did not reveal an altered methylation of *PITX2* in malignant cells, highlighting that hypermethylation of *PITX2* might be associated with a specific phenotype of PCa [51]. The exact mechanism by which *PITX2* promotes tumorigenesis is not fully understood; the data are limited to its regulatory role in the expression of the androgen receptor and insulin-like growth factor-I receptor [52].

Based on the single cell methylome analysis data in CAFs from patients with PCa [19], we expected that the methylation of *GATA6* would also reflect the abundance of CAF, which was not the case: the hypermethylation of *GATA6* was not associated with any CAF markers and other clinically relevant variables, unlike the hypermethylation of *PITX2*. This observation highlights that in a whole tissue specimen the overall methylation of *GATA6* might be impacted by the DNA of some other present cells as it probably is in the case of the *PITX2* methylation. However, there is no evidence of GATA6 hypomethylation in PCa cells [51]. Of note, our ddPCR assay for *GATA6* was more susceptible to the “rain effect” than others, which might have influenced the accuracy of the interpretation of its results [23].

Our results regarding the prognostic significance of the *EDARADD* hypomethylation in PCa patients confirm the findings of Lawrence et al. who examined methylation profiles in several published datasets [19]. The same observation was made in two recent studies which focused on CAF-specific methylation across various oncological diseases (lung, esophageal, and gastric cancers) [53,54]. Moreover, Shahabi et al. demonstrated that the expression of this gene is upregulated in PCa tissue from patients with a recurrent disease [55].

*EDARADD* is coding an adaptor protein that mediates the intracellular signaling of the extracellular Ectodysplasin A receptor (EDAR); this signaling is associated with various biological outcomes, including apoptosis, proliferation, and differentiation [56]. This pathway plays a key role in the development of ectodermal tissues (hair, teeth, etc.), and mutations in *EDARADD* are known to cause ectodermal malformations [57]. *EDARADD* is often included into gene panels for epigenetic clocks, as its methylation tends to decrease with time in various tissues, including the prostate [19,58,59,60]. In our cohort, methylation levels of *EDARADD* negatively correlated with methylation levels of both *PITX2* and *GATA6*, and its prognostic performance was strengthened by the addition of the PDGFRb surface expression into the model, which highlights that CAFs might be responsible for the hypomethylated fraction of this gene in the analyzed samples. As there is no evidence of a biological link between Ectodysplasin A receptor and PDGFR pathways, the combined prognostic significance of the *EDARADD* methylation and PDGFRb surface expression might be explained by the ability of the latter to reflect the abundance of CAFs, whereas the former is related to the behavior of these cells. Even though the role of *EDARADD* in the development and aggressiveness of TME components is still unknown, it is possible that it corresponds to an aged phenotype of CAFs, which leads to aberrant interactions between the TME and surrounding epithelium. Future studies on the role of *EDARADD* in the formation and function of the TME are warranted.

This study had several limitations. To begin, the cohort was rather small, and we were unable to collect data regarding the outcome of PCa for some patients, which could negatively influence the overall significance of the results and increase the risk of model overfitting. The developed prognostic model was not cross-validated in a different cohort, which narrows its translational potential. However, it is worth noting that all statistical tests implemented in this study had sufficient power at α = 0.05 and β = 0.2 for the respective sample sizes. Next, this study did not include samples of benign prostate hyperplasia or normal prostate tissue, thus the diagnostic potential of the studied CAF biomarkers remains unknown. Finally, the median follow-up time was 44 months, which does not encompass the full spectrum of PCa recurrence expected in clinical settings. The limitations stated above highlight the exploratory nature of this study. Translation of its results into clinical practice requires further research.

## 5. Conclusions

CAFs are emerging as a prominent biological substrate to investigate the intricacies of tumorigenesis. Many aspects of their involvement in cancer growth and invasion into surrounding tissues have not yet been discovered, but there is no doubt that they are one of the key mediators of these processes. Our study revealed the relationships between the CAF-specific DNA methylation, their expression of surface markers, and various clinical and morphological features of patients with PCa. Importantly, we identified the hypomethylation of *EDARADD* and increased surface expression of PDGFRb in tumor tissue samples obtained through radical treatment as promising stromal biomarkers of the biochemical recurrence of PCa. Upon further research and validation, the proposed model might become useful in the development of individualized follow-up strategies and identification of patients requiring more aggressive treatment options after surgery.

## Figures and Tables

**Figure 1 diagnostics-15-02434-f001:**
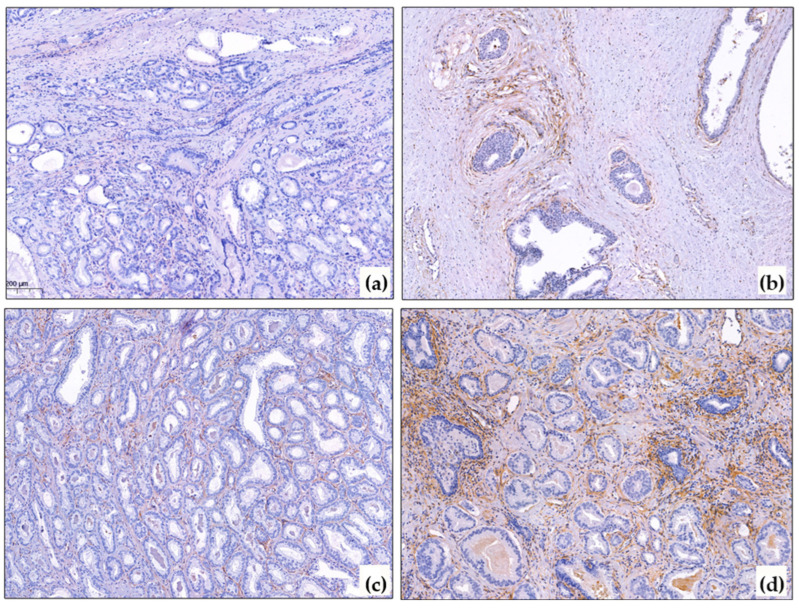
PDGFRb immunohistochemical reaction in PCa. (**a**) Negative PDGFRb reaction in cancer and around non-tumorous glands, ×10; (**b**) Positive PDGFRb expression around basal-cell hyperplasia, ×10; (**c**) Mild PDGFRb reaction in prostate cancer, ×10; (**d**) Strong PDGFRb expression in prostate cancer, ×10. The blue color corresponds to hematoxylin, which was used to stain the cell nuclei after the immunohistochemistry reaction. The brown color corresponds to DAB-chromogen, which was used in the immunohistochemical reaction.

**Figure 2 diagnostics-15-02434-f002:**
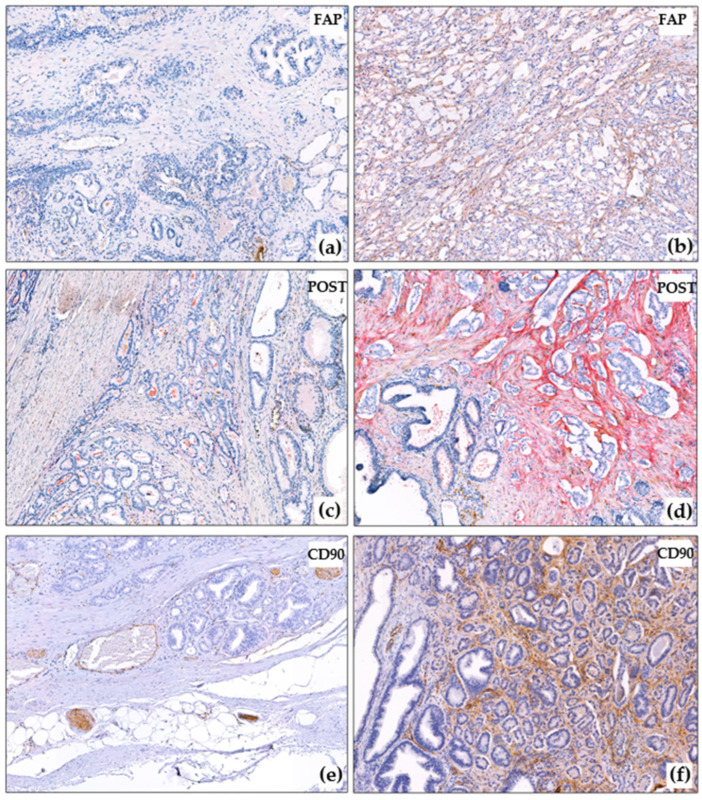
Immunohistochemical reactions in PCa. (**a**) Negative FAP reaction in cancer and around non-tumorous glands, ×10; (**b**) Strong FAP expression in prostate cancer, ×10; (**c**) Negative POST reaction in cancer and around non-tumorous glands, ×10; (**d**) Strong POST expression in prostate cancer, negative reaction around intact glands, ×10; (**e**) Negative CD90 reaction in prostate cancer, positive reaction in nerves and capillaries, ×10; (**f**) Strong CD90 expression in prostate cancer, negative reaction around intact glands, ×10. The blue color corresponds to hematoxylin, which was used to stain the cell nuclei after the immunohistochemistry reaction. The brown color corresponds to DAB-chromogen, which was used in the immunohistochemical reaction. The red color correposnds to Fast Red chromogen, which was used for the detection of POST.

**Figure 3 diagnostics-15-02434-f003:**
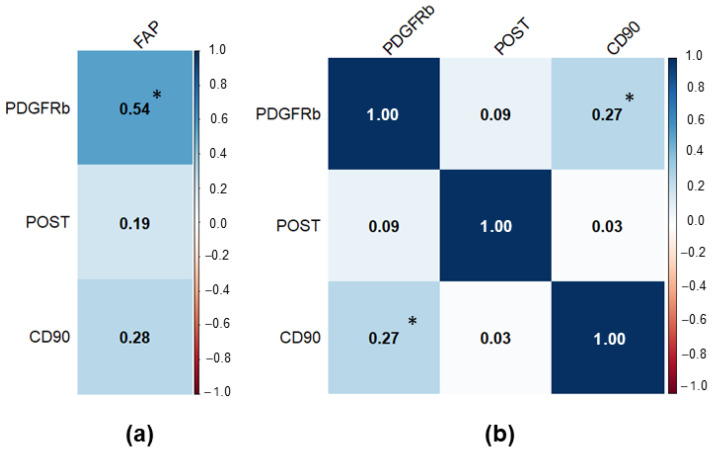
Correlation matrix for the studied CAF markers in PCa samples. Significant results at *p* < 0.05 were marked with “*”. (**a**) Correlation matrix for FAP (separate due to low sample size); (**b**) Correlation matrix for PDGFRb, POST, and CD90.

**Figure 4 diagnostics-15-02434-f004:**
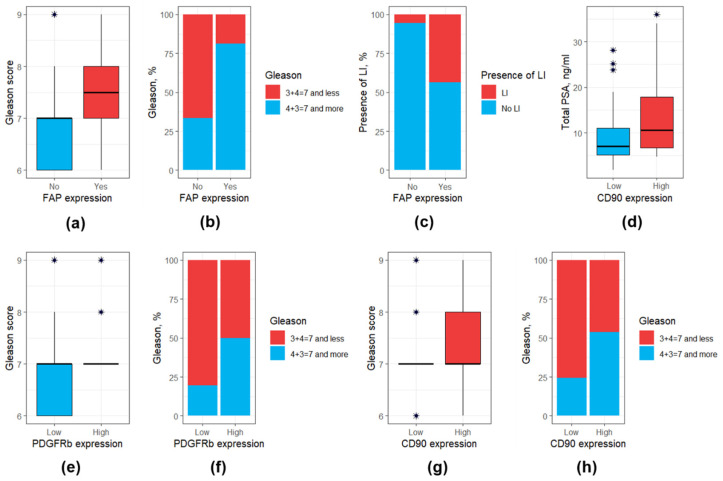
Box and bar plots for clinical and morphological features differing significantly (at *p* < 0.05) based on the expression of CAF surface markers. (**a**) FAP expression: Gleason score (values); (**b**) FAP expression: Gleason score (distribution); (**c**) FAP expression: perilymphatic invasion; (**d**) CD90 expression: total PSA; (**e**) PDGFRb expression: Gleason score (values); (**f**) PDGFRb expression: Gleason score (distribution); (**g**) CD90 expression: Gleason score (values); (**h**) CD90 expression: Gleason score (distribution). Outliers were marked with a “*” sign. PSA, prostate-specific antigen.

**Figure 5 diagnostics-15-02434-f005:**
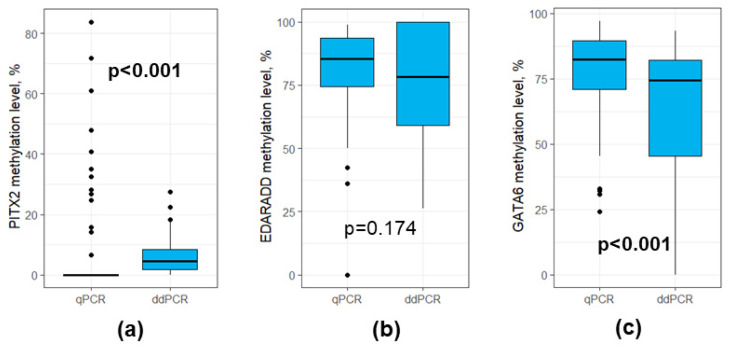
Boxplots for qPCR and ddPCR results for DNA methylation levels (**a**) *PITX2*; (**b**) *EDARADD*; (**c**) *GATA6*. Outliers were denoted using black dots.

**Figure 6 diagnostics-15-02434-f006:**
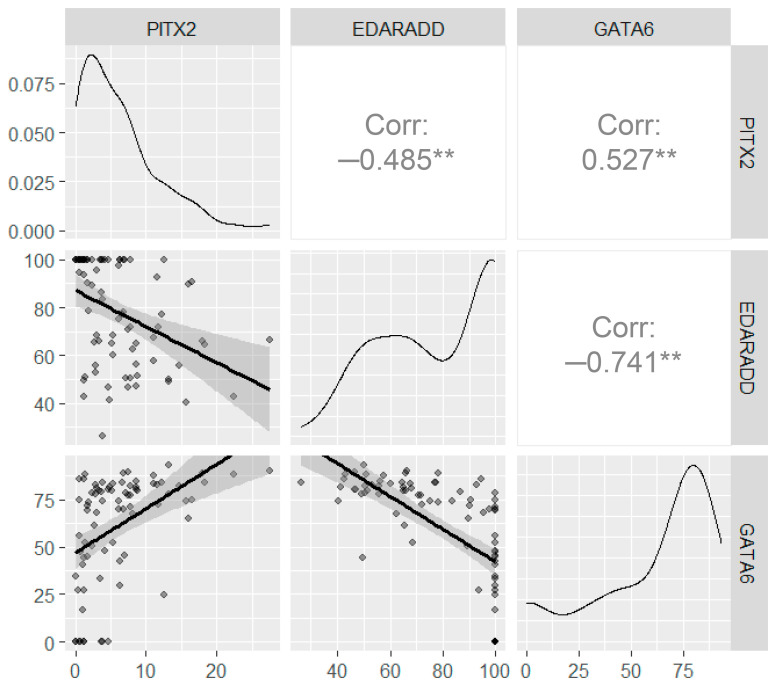
Correlation analysis for the DNA methylation levels assessed using ddPCR. The upper right panels show the correlation between the DNA methylation, the bottom left panels show the scatter plots, the diagonal panels show the density plots. All axes on the diagrams correspond to methylation levels (%). Significance levels below 0.001 are marked with “**”.

**Figure 7 diagnostics-15-02434-f007:**
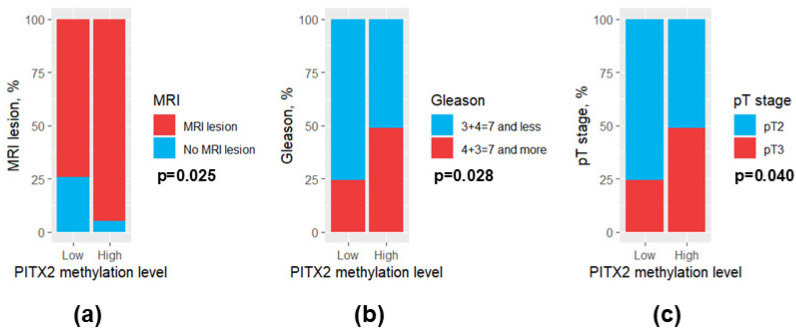
Bar plots for clinical and morphological features differing significantly between patients with low and high *PITX2* methylation levels. (**a**) MRI lesion; (**b**) Gleason score; (**c**) pT stage. MRI, magnetic resonance imaging.

**Figure 8 diagnostics-15-02434-f008:**
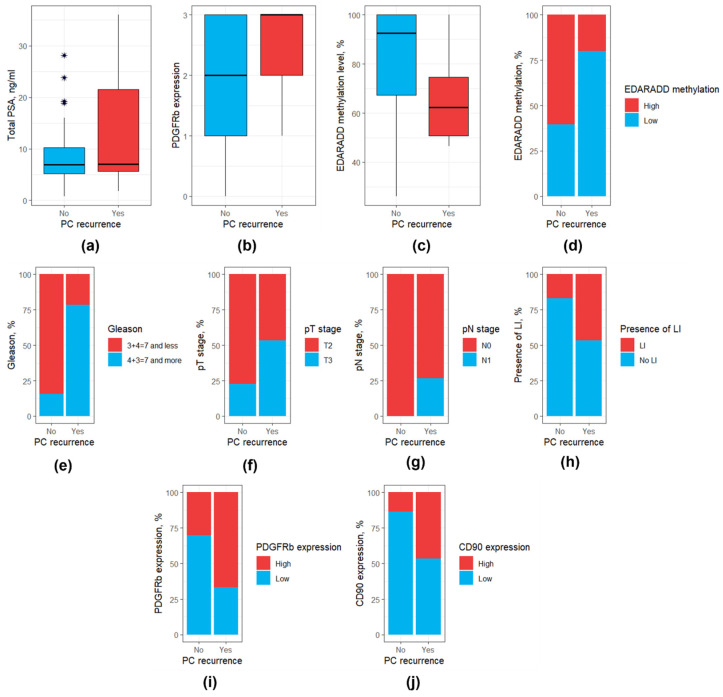
Box and bar plots for parameters differing significantly between patients with and without PCa recurrence. (**a**) Total PSA level; (**b**) PDGFRb expression (as a continuous variable); (**c**) *EDARADD* methylation level (as a continuous variable); (**d**) *EDARADD* methylation level (as a categorical variable); (**e**) Gleason score; (**f**) pT stage; (**g**) pN stage; (**h**) Presence of perilymphatic invasion. (**i**) PDGFRb expression (as a categorical variable); (**j**) CD90 expression (as a categorical variable). PSA, prostate specific antigen; LI, perilymphatic invasion. Outliers were marked with a “*” sign.

**Figure 9 diagnostics-15-02434-f009:**
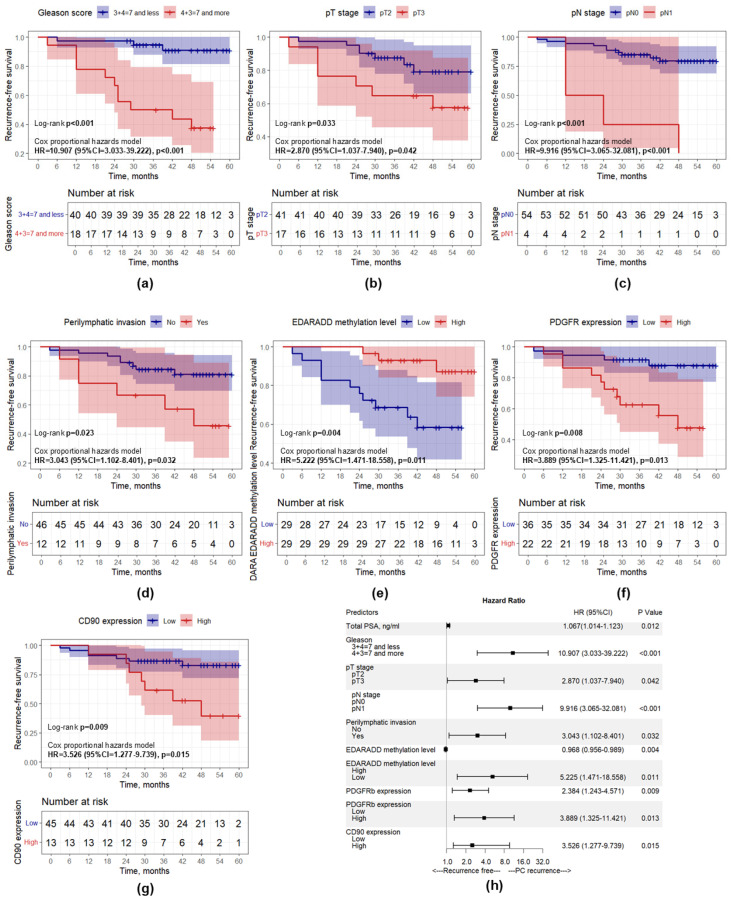
Kaplan–Meier curves showing recurrence-free survival differences. (**a**) Gleason score; (**b**) pT stage; (**c**) pN stage; (**d**) Perilymphatic invasion; (**e**) *EDARADD* methylation level; (**f**) PDGFRb expression; (**g**) CD90 expression; (**h**) Forest plot representing the results of univariate Cox regression analysis.

**Figure 10 diagnostics-15-02434-f010:**
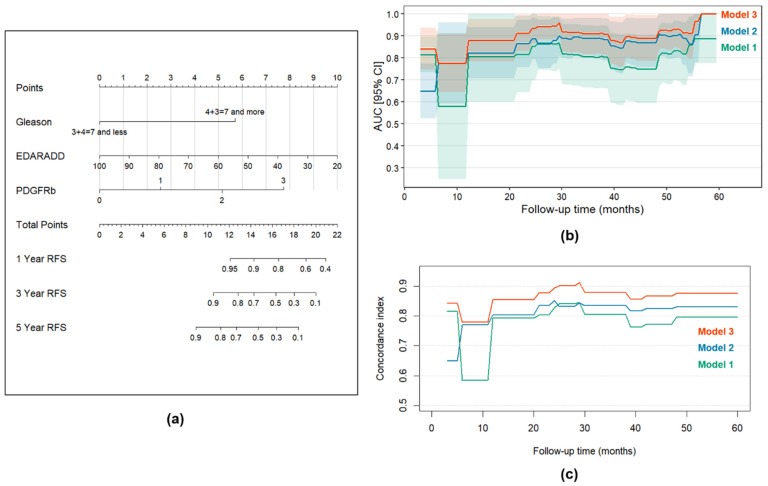
Time-dependent prediction of survival. (**a**) Nomogram for the assessment of recurrence-free survival in patients with PCa at 1, 3, and 5 years after the surgery for *Model 3*; (**b**) Time-dependent AUC values for *Models 1, 2 and 3* with ROC-AUCs for 60 months follow-up for every 1-month timepoint; (**c**) C-indices for *Models 1, 2 and 3* during a 5-year follow-up period. The C-indices reflect how well patients are sorted according to the recurrence rate. Shadowed areas correspond to 95% confidence intervals.

**Table 1 diagnostics-15-02434-t001:** Demographic and clinical characteristics of the study participants.

Characteristics	*n* = 88
Age, median (Q1–Q3)	64.0 (60.0–68.5)
BMI, kg/m^2^, median (Q1–Q3)	27.2 (25.4–29.8)
PSA, ng/mL, median (Q1–Q3)	7.1 (5.2–13.6)
MRI lesion, % (*n*)	84.1% (74)
Prostate volume, cm^3^, median (Q1–Q3)	36.1 (30.0–51.5)
Gleason	
3 + 3 = 6	21.2% (18)
3 + 4 = 7	45.9% (39)
3 + 5 = 8	1.2% (1)
4 + 3 = 7	17.6% (15)
4 + 4 = 8	9.4% (8)
4 + 5 = 9	3.5% (3)
5 + 4 = 9	1.2% (1)
Androgen-deprivation therapy	3.4% (3)
pT stage	
pT2, % (*n*)	68.2% (60)
pT3, % (*n*)	31.8% (28)
pN stage, % (*n*)	
0, % (*n*)	91.9% (79)
1, % (*n*)	8.1% (7)
Biochemical recurrence, % (*n*) ^1^	21.7% (15) ^2^

^1^ Within an up to 5-year observation period. ^2^ Data for 19/88 patients were not available. BMI, body mass index; PSA, prostate-specific antigen; MRI, magnetic resonance imaging.

**Table 2 diagnostics-15-02434-t002:** Expression of CAF markers in prostate cancer specimens (semi-quantitative scale).

Scale	PDGFRb	FAP	POST	CD90
0 (negative)	10 (11.4%)	18 (52.9%)	10 (11.4%)	53 (61.6%)
1 (mild)	25 (28.4%)	9 (26.5%)	21 (23.9%)	13 (15.1%)
2 (moderate)	22 (25.0%)	6 (17.7%)	20 (22.7%)	7 (8.2%)
3 (strong)	31 (35.2%)	1 (2.9%)	37 (42.0%)	13 (15.1%)
Total	88	34 *	88	86 *

* A portion of samples was not included due to insufficient material. Data were obtained based on the immunohistochemical analysis results.

**Table 3 diagnostics-15-02434-t003:** Expression of CAF markers in prostate cancer specimens (qualitative scale).

	FAP		PDGFRb	POST	CD90
Negative	18 (52.9%)	Low expression	57 (64.8%)	31 (35.2%)	66 (76.7%)
Positive	16 (47.1%)	High expression	31 (35.2%)	57 (64.8%)	20 (23.3%)
Total	34 *		88	88	86 *

* A portion of samples was not included due to insufficient material. Data were obtained based on the immunohistochemical analysis results.

**Table 4 diagnostics-15-02434-t004:** Comparison of clinical and morphological parameters, DNA methylation levels, and CAF markers between patients with and without prostate cancer recurrence.

Parameter	No Recurrence *n* = 53	Recurrence *n* = 15	*p*
Age, years, median (Q1–Q3)	65.0 (60.0–68.0)	64.0 (60.5–68.0)	0.999
BMI, kg/m^2^, median (Q1–Q3)	27.1 (25.1–29.8)	26.4 (25.3–27.4)	0.546
PSA, ng/mL, median (Q1–Q3)	6.9 (5.2–10.3)	7.0 (5.7–21.5)	0.009 *
MRI lesion, % (*n*)	83.0% (44)	80.0% (12)	0.719
Prostate volume, cm^3^, median (Q1–Q3)	36.0 (28.3–59.0)	36.0 (30.0–43.0)	0.464
Gleason			<0.001 *
3 + 4 = 7 and less, % (*n*)	84.3% (43)	21.4% (3)	
4 + 3 = 7 and more, % (*n*)	15.7% (8)	78.6% (11)	
pT stage			0.029 *
pT2, % (*n*)	77.4% (41)	46.7% (7)	
pT3, % (*n*)	22.6% (12)	53.3% (8)	
pN stage, % (*n*)			0.002 *
0, % (*n*)	100.0% (51)	73.3% (11)	
1, % (*n*)	0.0% (0)	26.7% (4)	
Pn, % (*n*)	84.9% (45)	80.0% (12)	0.696
LI, % (*n*)	17.0% (9)	46.7% (7)	0.034 *
*PITX2* methylation level, median (Q1–Q3)	3.8 (1.7–7.8)	7.5 (3.5–10.3)	0.098
*PITX2* methylation level			0.154
Low, % (*n*)	52.1% (25)	26.7% (4)	
High, % (*n*)	47.9% (23)	73.3% (11)	
*EDARADD* methylation level, median (Q1–Q3)	92.6 (67.3–100.0)	62.4 (50.8–74.6)	0.004 *
*EDARADD* methylation level			0.015 *
Low, % (*n*)	39.6% (19)	80.0% (12)	
High, % (*n*)	60.4% (29)	20.0% (3)	
*GATA6* methylation level, median (Q1–Q3)	70.5 (40.3–79.9)	82.0 (73.0–86.3)	0.064
*GATA6* methylation level			0.163
Low, % (*n*)	58.3% (28)	33.3% (5)	
High, % (*n*)	41.7% (20)	66.7% (10)	
PDGFRb, median (Q1–Q3)	2.0 (1.0–3.0)	3.0 (2.0–3.0)	0.005 *
PDGFRb			0.023 *
Low, % (*n*)	69.8% (37)	33.3% (5)	
High, % (*n*)	30.2% (16)	66.7% (10)	
FAP, median (Q1–Q3)	0.0 (0.0–0.8)	1.0 (0.3–1.8)	0.084
FAP			0.179
No, % (*n*)	70.0% (10)	30.0% (3)	
Yes, % (*n*)	30.0% (3)	70.0% (7)	
POST, median (Q1–Q3)	3.0 (1.0–3.0)	2.0 (1.0–2.0)	0.113
POST			0.745
Low, % (*n*)	26.4% (14)	33.3% (5)	
High, % (*n*)	73.6% (39)	66.7% (10)	
CD90, median (Q1–Q3)	0.0 (0.0–1.0)	0.0 (0.0–1.0)	0.404
CD90			0.010 *
Low, % (*n*)	86.5% (45)	53.3% (8)	
High, % (*n*)	13.5% (7)	46.7% (7)	

BMI, body mass index; PSA, prostate-specific antigen; MRI, magnetic resonance imaging. Significance levels below 0.05 are marked with “*”.

**Table 5 diagnostics-15-02434-t005:** Univariate and multivariate Cox regression analyses for prostate cancer recurrence.

Factors	HR (95% CI)	*p*	HR (95% CI)	*p*
Univariate			Multivariate	
**Clinical and morphological predictors**			***Model 1* (AIC = 94.6)**	
Total PSA, ng/mL	1.067 (1.014–1.123)	0.012 *	-	
Gleason				
3 + 4 = 7 and less	Reference		Reference	
4 + 3 = 7 and more	10.907 (3.033–39.222)	<0.001 *	8.013 (2.069–31.041)	0.003 *
pT stage			-	
pT2	Reference			
pT3	2.870 (1.037–7.940)	0.042 *		
pN stage				
0	Reference		Reference	
1	9.916 (3.065–32.081)	<0.001 *	3.371 (0.964–11.787)	0.057
LI	3.043 (1.102–8.401)	0.032 *	-	
**New predictors**			***Model 2* (AIC = 103.1)**	
*EDARADD* methylation level	0.968 (0.956–0.989)	0.004 *	0.963 (0.939–0.987)	0.003 *
*EDARADD* methylation level				
Low	5.225 (1.471–18.558)	0.011 *	-	
High	Reference			
PDGFRb expression	2.384 (1.243–4.571)	0.009 *	2.571 (1.340–4.935)	0.005 *
PDGFRb expression				
Low	Reference		-	
High	3.889 (1.325–11.421)	0.013 *		
CD90 expression		0.015 *	-	
Low	Reference			
High	3.526 (1.277–9.739)			

AIC, Akaike information criterion; PSA, prostate specific antigen; LI, perilymphatic invasion. Significance levels below 0.05 are marked with “*”.

**Table 6 diagnostics-15-02434-t006:** A mixed Cox regression analysis for predicting the risk of PCa recurrence.

*Model 3* (AIC = 87.0)		
Factors	HR (95% CI)	*p*
Gleason		
3 + 4 = 7 and less	Reference	
4 + 3 = 7 and more	6.247 (1.627–23.988)	0.008 *
*EDARADD* methylation level	0.961 (0.931–0.991)	0.013 *
PDGFRb expression	2.313 (1.054–5.088)	0.036 *

Significance levels below 0.05 are marked with “*”.

## Data Availability

All data generated in the present study is available in Appendix A.

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
