# Peer review of "Prognostic Potential of Cancer-Associated Fibroblast Surface Markers and Their Specific DNA Methylation in Prostate Cancer"

_diagnostics, 2025, doi:10.3390/diagnostics15192434_

Round 1

Reviewer 1 Report

Comments and Suggestions for Authors

Overall, this is a well-written and presented study. The authors evaluate several potential  biomarkers / DNA methylation sites for prognostic value. In addition, qPCR was compared to ddPCR within the analysis.

With the patient characteristics table (Table 1), the patients are not stratified by recurrence versus no recurrence. Since comparing these two groups is a focus of this study, the patients should be stratified by these two groups in this table. Or at the very least, it should be included as a characteristic of the population. But it really should be stratified to assess for differences, like done in table 4.  Then Table 4 could focus on the markers and methylation analyses.  Likewise, in Tables 2 & 3, the marker information should also be stratified into these two groups. This is important for visualization / interpretation of the data provided.  

Why weren’t normal prostate stromal cells analyzed? This would be an important component to evaluate the specificity of the biomarker use. Comparison to stroma in BPH only samples would also provide further validation.

As the authors acknowledge in the discussion, it is a concern is that there is only an n=15 with recurrence, and the total number of samples overall, is a rather low.  Adding a line or two as to how large of a cohort would be appropriate for validation of the model would be good to emphasize this.

Minor issue: The dark blue background color on three of the boxes in figure 3 makes it too dark / difficult to read the numbers in the box.  Even though there are all values of 1.00, using a white font for the text, or a lighter background color, needs to be used for visibility.

Author Response

Esteemed Reviewer,

We are grateful for the time and effort dedicated to providing valuable comments on our manuscript. We have done our best to revise the manuscript and include all the suggestions provided. The changes are highlighted in the revised manuscript using MS Word tools.

Below you may find a point-by-point response to received comments.

Comment 1: With the patient characteristics table (Table 1), the patients are not stratified by recurrence versus no recurrence. Since comparing these two groups is a focus of this study, the patients should be stratified by these two groups in this table. Or at the very least, it should be included as a characteristic of the population. But it really should be stratified to assess for differences, like done in table 4.  Then Table 4 could focus on the markers and methylation analyses.  Likewise, in Tables 2 & 3, the marker information should also be stratified into these two groups. This is important for visualization / interpretation of the data provided.

Response: Thank you for this comment. We agree that the table with clinical and demographic characteristics of the study participants is missing information regarding the recurrence rate. This row has been added to Table 1. Our team deliberately chose to separate the descriptive data from the comparative/regression data within the text. If we include differences between patients with and without recurrence in the materials and methods, we will also have to report the corresponding p-values (otherwise, it would not be as informative). If we include this information, we will have to comment on it in the corresponding section of the results, which is 5+ pages ahead. The same is true for the stratification of the data in Tables 2 and 3. Subsection 3.4 of the results is dedicated to the prediction of PCa recurrence. This subsection includes the suggested by the Reviewer stratification of all relevant biomarker-related, clinical and demographic data (for patients with and without the recurrence) with corresponding p-values for significantly differing variables. These data are then used to create 3 predictive models (please note, that we used not only IHC and methylation markers here, but some clinical data as well). Our team believes that if these data were transferred to previous sections of the text, it would be inconvenient for the reader, thus disrupting the flow of the manuscript. We hope that the Reviewer finds our explanation of the decision to not alter these tables reasonable and just.

Comment 2: Why weren’t normal prostate stromal cells analyzed? This would be an important component to evaluate the specificity of the biomarker use. Comparison to stroma in BPH only samples would also provide further validation.

Response: We appreciate this suggestion. Our study was dedicated to the evaluation of the prognostic potential of several IHC markers and methylation in PCa CAFs. We did not focus on the diagnostic/classification usage of these analytes. Therefore, our methodology did not include a dedicated analysis of the specificity in normal/BPH tissue. In fact, initially, we included 19 BPH samples, but the amount of material was abysmal as it was obtained at prostate biopsy, and the whole samples were not available for both IHC and methylation analysis after the routine analysis at the time of inclusion. Results of the analysis of these samples did not meet our quality control criteria declared in the materials and methods section. Therefore, these samples were not included in the study. Nevertheless, our study includes staining of normal prostate and hyperplasia tissue within the same samples. In the beginning of the subsection 3.1 “CAF markers: immunohistochemistry” we describe the pattern of the presence of these markers not only in the tumor tissue but also around it (Figures 1-2). In the beginning of the discussion section we present the data from various sources, which highlights that these markers are not expressed in fibroblasts from cancer-free tissue. As for the methylation analysis, specificity was assessed only during the preclinical validation in totally methylated/unmethylated Qiagen control DNA, which is stated in the materials and methods section.

To address the issue raised by the Reviewer, we have expanded the study limitations paragraph to include the following statement:
“…Next, this study did not include samples of benign prostate hyperplasia or normal prostate tissue, thus the diagnostic potential of the studied CAF biomarkers remains unknown…”

Comment 3: As the authors acknowledge in the discussion, it is a concern is that there is only an n=15 with recurrence, and the total number of samples overall, is a rather low.  Adding a line or two as to how large of a cohort would be appropriate for validation of the model would be good to emphasize this.

Response: We are grateful for this comment. According to Eric Vittinghoff and Charles E. McCulloch (https://doi.org/10.1093/aje/kwk052), the rule of thumb that logistic and Cox models should be used with a minimum of 10 outcome events per predictor variable based on two simulation studies, may be too conservative. They conclude that this rule can be relaxed (to a point of 1 factor per 5 outcomes), in particular for sensitivity analyses undertaken to demonstrate adequate control of confounding. However, we acknowledge that in our study the number of recurrence events was quite low, as is stated in the study limitations section. We have changed the wording slightly to better emphasize this issue. Moreover, we have added internal validation at the end of the results section and explicitly acknowledged the need for external validation.

“…Additionally, Model 3 was validated internally using bootstrap (1000 replications): the Dxy was 0.727, whereas Harrell's C-index was 0.864, confirming the discrimination ability of the proposed model.”

“To begin, the cohort was rather small, and we were unable to collect data regarding the outcome of PCa for some patients, which could negatively influence the overall significance of the results and increase the risk of model overfitting. The developed prognostic model was not cross-validated in a different cohort, which narrows its translational potential. However, it is worth noting that all statistical tests implemented in this study had sufficient power at α=0.05 and β=0.2 for the respective sample sizes…”

 Comment 4: Minor issue: The dark blue background color on three of the boxes in figure 3 makes it too dark / difficult to read the numbers in the box.  Even though there are all values of 1.00, using a white font for the text, or a lighter background color, needs to be used for visibility.

Response: Thank you for pointing this out. Figure 3 was modified accordingly to ensure better visibility.

Reviewer 2 Report

Comments and Suggestions for Authors

  1. Sample size / model robustness
    • The number of recurrence events (n=15) is low relative to the number of predictors used. Please discuss explicitly the risk of overfitting and optimism bias in your multivariable models.
  2. Validation
    • The reported AUC of 1.0 at 5 years seems overly optimistic. Consider adding internal validation (e.g., bootstrapping) or explicitly acknowledge the need for external validation in future cohorts.
  3. Follow-up duration
    • Median follow-up (~25 months) may not capture the full spectrum of prostate cancer recurrence. Please comment on how this limitation might affect prognostic conclusions, particularly beyond 3 years.
  4. IHC reproducibility
    • Provide details on how PDGFRβ and CD90 staining were scored (number of observers, interobserver variability). This will improve transparency regarding reproducibility.
  5. Feasibility of ddPCR
    • Expand on the clinical applicability of ddPCR: costs, availability in pathology labs, and potential barriers to routine implementation.
  6. Biological rationale
    • Please elaborate on potential biological links between CAF activation (PDGFRβ) and EDARADD methylation. Are these independent prognostic pathways or mechanistically connected?
  7. Results clarity
    • Clarify why PITX2, despite association with adverse features, did not remain significant in multivariable recurrence analyses.
    • Consider harmonizing decimal places and layout in tables for readability.
  8. Abstract / conclusions
    • Temper conclusions to reflect that findings suggest prognostic potential rather than implying immediate clinical utility.

Author Response

Esteemed Reviewer,

We are grateful for the time and effort dedicated to providing valuable comments on our manuscript. We have done our best to revise the manuscript and include all the suggestions provided. The changes are highlighted in the revised manuscript using MS Word tools.

Below you may find a point-by-point response to received comments.

Comment 1: The number of recurrence events (n=15) is low relative to the number of predictors used. Please discuss explicitly the risk of overfitting and optimism bias in your multivariable models.

Response: We are grateful for this comment. According to Eric Vittinghoff and Charles E. McCulloch (https://doi.org/10.1093/aje/kwk052), the rule of thumb that logistic and Cox models should be used with a minimum of 10 outcome events per predictor variable based on two simulation studies, may be too conservative. They conclude that this rule can be relaxed (to a point of 1 factor per 5 outcomes), in particular for sensitivity analyses undertaken to demonstrate adequate control of confounding. However, we acknowledge that in our study the number of recurrence events was quite low, as is stated in the study limitations section. We have changed the wording slightly to better emphasize this issue.

“To begin, the cohort was rather small, and we were unable to collect data regarding the outcome of PCa for some patients, which could negatively influence the overall significance of the results and increase the risk of model overfitting. The developed prognostic model was not cross-validated in a different cohort, which narrows its translational potential. However, it is worth noting that all statistical tests implemented in this study had sufficient power at α=0.05 and β=0.2 for the respective sample sizes…”

Comment 2: The reported AUC of 1.0 at 5 years seems overly optimistic. Consider adding internal validation (e.g., bootstrapping) or explicitly acknowledge the need for external validation in future cohorts.

Response: Thank you for pointing this out. We have added internal validation at the end of the results section. And we have explicitly acknowledged the need for external validation (please see the response to Comment 1 above).

“…Additionally, Model 3 was validated internally using bootstrap (1000 replications): the Dxy was 0.727, whereas Harrell's C-index was 0.864, confirming the discrimination ability of the proposed model.”

Comment 3: Median follow-up (~25 months) may not capture the full spectrum of prostate cancer recurrence. Please comment on how this limitation might affect prognostic conclusions, particularly beyond 3 years.

Response: Thank you for this comment. It is worth noting that 25 months was not the median follow-up time for our cohort. It was the median time to recurrence. As it is stated in subsection 2.1 of the Materials and Methods, the follow-up time was 3-6 years. According to Table S1, the median follow-up time was close to 4 years. We agree that this still does not capture the full spectrum of PCa recurrence, thus the following line has been added to the study limitations section:

“…Finally, the median follow-up time was 44 months, which does not encompass the full spectrum of PCa recurrence expected in clinical settings.”

Comment 4: Provide details on how PDGFRβ and CD90 staining were scored (number of observers, interobserver variability). This will improve transparency regarding reproducibility.

Response: We appreciate this suggestion. Materials and methods section has been modified accordingly (see below).

“In prostate cancer, PDGFRb was evaluated around tumor complexes. The intensity of the reaction was assessed in the designated area using a scale of 0 to 3 (0 – no reaction, 1 – weak reaction, 2 – moderate reaction, 3 – strong reaction). FAP and POST were evaluated using the same scale. CD90 showed a very heterogeneous and mosaic pattern, so this re-action was evaluated as a percentage of the stroma. Subsequently, the percentage was converted to a standard system (0-3) by analogy with other CAF markers as follows: 0-6% − negative; 7-15% − 1, 16-39 − 2, >40% − 3. The average staining intensities from 5 fields of vision were counted for each sample. The quantitative assessment of the stained micro-graphs was performed independently by two pathologists to achieve a more objective value. In case of discrepancy between observations, the mean value was calculated.”

Comment 5: Expand on the clinical applicability of ddPCR: costs, availability in pathology labs, and potential barriers to routine implementation.

Response: The corresponding segment of the discussion section has been expanded to include the suggested information (see below).

“The advantages of ddPCR might be outweighed by the prices of the instruments, reagents, and consumables. ddPCR systems are 5-10 more expensive than those for qPCR, and the price of consumables per reaction is higher by approximately the same margin, limiting the availability of these systems across pathology laboratories. The medico-economic benefit of its translation into clinical practice for the purposes of the prognosis evaluation in PCa needs to be investigated further.”

Comment 6: Please elaborate on potential biological links between CAF activation (PDGFRβ) and EDARADD methylation. Are these independent prognostic pathways or mechanistically connected?

Response: The corresponding segment of the discussion section has been expanded to include the suggested information (see below).

“…This pathway plays a key role in the development of ectodermal tissues (hair, teeth, etc.), and mutations in EDARADD are known to cause ectodermal malformations [57]. EDARADD is often included into gene panels for epigenetic clocks, as its methylation tends to decrease with time in various tissues, including the prostate [19,58–60]. In our cohort, methylation levels of EDARADD negatively correlated with methylation levels of both PITX2 and GATA6, and its prognostic performance was strengthened by the addition of the PDGFRb surface expression into the model, which highlights that CAFs might be responsible for the hypomethylated fraction of this gene in the analyzed samples. As there is no evidence of a biological link between Ectodysplasin A receptor and PDGFR pathways, the combined prognostic significance of the EDARADD methylation and PDGFRb surface expression might be explained by the ability of the latter to reflect the abundance of CAFs, whereas the former is related to the behavior of these cells. Even though the role of EDARADD in the development and aggressiveness of TME components is still unknown, it is possible that it corresponds to an aged phenotype of CAFs, which leads to aberrant interactions between the TME and surrounding epithelium. Future studies on the role of EDARADD in the formation and function of the TME are warranted.”

Comment 7: Clarify why PITX2, despite association with adverse features, did not remain significant in multivariable recurrence analyses. Consider harmonizing decimal places and layout in tables for readability.

Response: We appreciate this suggestion. Decimal places were harmonized within each table. Lines 508-522 of the manuscript are dedicated to the discussion of our results related to the PITX2 methylation. We state that this outcome was unexpected for us based on the available literature (it was not previously studied using ddPCR, but in our case this biomarker was not significant for qPCR as well). The p-value for PITX2 methylation (regarding the recurrence) was 0.098. It could be said that there was a trend, it just did not reach the significance threshold. There is a chance that expansion of our cohort could render this analyte prognostically significant. This study was exploratory, so the sample size was indeed limited. But we do not want to hypothesize on this level. It is worth noting that previously this biomarker was not studied in the population of our country, but we do not have enough evidence that PCa epigenetics in our region are that different compared to European. Clinical characteristics of our cohort also do not differ from that in other studies. Therefore, we have a hard time providing any additional information to Lines 508-522 of the discussion without it being speculative. Based on positive results of our study (Model 3), we have decided to continue the research in this field. We hope that our future data derived from a larger cohort will allow us to get a better insight into the prognostic role of this biomarker.

Comment 8: Temper conclusions to reflect that findings suggest prognostic potential rather than implying immediate clinical utility.

Response: Thank you for pointing this out. We have tempered the conclusions in the abstract so that they now correspond to the statements in the conclusions of the main text.

“Upon further validation, the abundance of CAFs and their specific methylation might become a promising tool for the assessment of prognosis in PCa after radical treatment.”

Round 2

Reviewer 2 Report

Comments and Suggestions for Authors

This is a well-written and timely study on CAF markers and DNA methylation in prostate cancer prognosis. The methodology and results are clear and promising. Minor clarifications are needed regarding DNA extraction and quality control, Cox regression models, and the interpretation of PITX2/GATA6 methylation. The limitations should also be stated more explicitly, and conclusions framed more cautiously. With these minor revisions, the manuscript will be suitable for publication.

Author Response

Esteemed Reviewer,

Thank you for pointing out the remaining drawbacks of our manuscript. We have done our best to revise the manuscript and include all the suggestions provided. The changes are highlighted in the revised manuscript using MS Word tools.

Below you may find a list of additions according to the review report.

  1. Comment 1: Minor clarifications for DNA extraction and quality control.
    Response: We have added the following information (DNA isolation and processing section of Materials and Methods):
    “The elution volume was set to 60 μL.”
    “The following quality control criteria were applied: DNA concentration >10 ng/μL and DNA purity (A260/A280) 1.7-1.9.”
  1. Comment 2: Minor clarifications for Cox regression models.
    Response: The following line has been added to the statistical analysis subsection of Materials and Methods:
    “Bootstrap was used for the internal validation of the developed models.”

  2. Comment 3: Minor additions to the interpretation of PITX2/GATA6 methylation
    Response: The following lines have been added to Discussion:
    “It is worth noting that the prognostic relevance of this biomarker was not studied previ-ously in the population of our region, and PCa is known to have significant regional and population-level genetic differences, which might explain this discrepancy [51].”
    “Interestingly, single cell methylome analysis of PCa, compared to a matching benign-appearing tissue, did not reveal an altered methylation of PITX2 in malignant cells, highlighting that hypermethylation of PITX2 might be associated with a specific phenotype of PCa [52].”
    “However, there is no evidence of GATA6 hypomethylation in PCa cells [52].”

  3. Comment 4: A more explicit statement of the study limitations.
    Response: The following lines have been added to the end of study limitations section of Discussion:
    “The limitations stated above highlight the exploratory nature of this study. Translation of its results into clinical practice requires further research.”

  4. Comment 5: More cautious framing of the conclusions.
    Response: The following line in the Conclusions has been modified to highlight that further research is warranted:
    “Upon further research and validation, the proposed model might become useful in the development of individualized follow-up strategies and identification of patients requiring more aggressive treatment options after the surgery”.